# Effects of *Auricularia auricula* Polysaccharides on Gut Microbiota and Metabolic Phenotype in Mice

**DOI:** 10.3390/foods11172700

**Published:** 2022-09-04

**Authors:** Qian Liu, Xin An, Yuan Chen, Yuxuan Deng, Haili Niu, Ruisen Ma, Haoan Zhao, Wei Cao, Xiaoru Wang, Meng Wang

**Affiliations:** 1College of Food Science and Technology, Northwest University, 229 Taibai North Road, Xi’an 710069, China; 2Shaanxi Functional Food Engineering Center Co., Ltd., Xi’an 710069, China

**Keywords:** *Auricularia auricula* polysaccharides, metabolomics, gut microbiota, differential metabolites, metabolic phenotype

## Abstract

Personalized diets change the internal metabolism of organisms, which, in turn, affects the health of the body; this study was performed to explore the regulatory effects of polysaccharides extracted from *Auricularia auricula* on the overall metabolism and gut microbiota in normal C57BL/6J mice. The study was conducted using metabolomic and microbiomic methods to provide a scientific basis for further development and use of *Auricularia auricula* resources in the Qinba Mountains and in nutritional food with *Auricularia auricula* polysaccharides (AAP) as the main functional component. Based on LC-MS/MS metabolomic results, 51 AAP-regulated metabolites were found, mainly enriched in the arginine biosynthesis pathway, which had the highest correlation, followed by the following metabolisms: arginine and proline; glycine, serine and threonine; and glycerophospholipid, along with the sphingolipid metabolism pathway. Furthermore, supplementation of AAP significantly changed the composition of the mice intestinal flora. The relative abundance levels of *Lactobacillus johnsonii*, *Weissella cibaria*, *Kosakonia cowanii*, *Enterococcus faecalis*, *Bifidobacterium animalis* and *Bacteroides uniformis* were markedly up-regulated, while the relative abundance of *Firmicutes bacterium M10-2* was down-regulated. The bioactivities of AAP may be related to the regulatory effects of endogenous metabolism and gut microbiota composition.

## 1. Introduction

*Auricularia auricula-judae*, commonly known as black fungus, is a common delicious and healthy food and in China has the reputation of being the “meat in plain food” [1]. The Chinese first cultivated *Auricularia auricula* 4000 years ago during the time of Emperor Shennong; it was noted in Li Shizhen’s *Compendium of Materia Medica* during the Ming dynasty that *Auricularia auricula* “benefits *qi*, no hunger, be light and better in memory” [2]. As the saying goes “Little fungus, big industry”, the Shaanxi Qinba Mountains, with its dense forests, sufficient sunlight, humidity and large summer-to-autumn temperature variation is also suitable for *Auricularia auricula* [3]. Polysaccharides, one of the most important active components in the fruiting body of *Auricularia auricula*, have strong antioxidant, immune regulation, weight-reduction, and fat-decreasing properties [4,5]. Although many recent studies have focused on using *Auricularia auricula* extract to prevent and treat metabolic and other disorders and as a cooking ingredient, its effect and potential mechanism on a healthy body remain unclear.

Numerous studies have shown that personalized diets change the internal metabolism of organisms, which, in turn, affects the health of the body [6]. Metabolomics is an important part of system biology, because it reveals changes in metabolites and metabolic pathways caused by the stimulation of endogenous substances [7]. In recent years, many researchers have used metabolomics to investigate the effects of food intervention on health. Esteban-Fernandez et al. developed an ultra-performance liquid chromatography-time-of-flight mass spectrometry metabolomics method to analyze the effects of moderate red wine consumption on the urine of healthy volunteers [8]. Liu et al. studied the changes in exogenous and endogenous metabolites in the plasma and urine of female college students and female SD rats induced by cranberry extract [9,10]; these metabolites can be used as biomarkers for cranberry juice consumption and to explain their health-promoting properties.

Host metabolism is controlled by metabolic pathways regulated by their own genomes and metabolic processes regulated by microbial genomes [11]. The influence of acquired external environmental factors, especially diet, on the composition and structure of flora cannot be ignored [12]. Many studies have shown that dietary polysaccharides with different structures could regulate the composition of intestinal flora precisely [13,14]. Xu et al. found that *Auricularia auricula* polysaccharides (AAP) elevated intestinal flora diversity and optimized microbial composition and function in T2D mice, characterized by increased *Lactobacillus* and *Bacteroides* and decreased *Clostridium* and *Allobaculum* [15]; however, there are few studies on the regulatory effects of AAP on gut microbiota in normal mice, especially concerning AAP from the Qinba Mountains.

In this study, metabolomic and intestinal microbiome methods were used to study the effects of Qinba mountain AAP on the serum metabolites and intestinal flora of normal male mice, and to reveal their regulatory effects on the overall metabolism and gut microbiota homeostasis of mammals.

## 2. Materials and Methods

### 2.1. Preparation of AAP

Thirty grams of *Auricularia auricula-judae* powder, purchased from Shaanxi Tianmei Green Industry Co., Ltd. (Hanzhong, China), and grown in the Qinba Mountains of northwest China, were weighed, and extracted using 100 °C water (solid–liquid ratio 1:50) for 2 h. The solution was centrifuged at 10,000 r/min for 10 min, and the supernatant was concentrated. Then, ethanol was added at a ratio of 1:4 (volume fraction), and alcohol precipitation was carried out for 12 h. The precipitates were separated under 10,000 r/min for 10 min, and fully dissolved in distilled water. After removing protein 4–5 times though the sevage (dichloromethane: n-butanol = 4:1, *v*/*v*) method, cellophane, with a molecular cut-off of 3000 Da, was used for dialysis in flowing tap water for 3 consecutive days, after which the solution was concentrated, freeze-dried, and bagged to obtain the AAP [4] (the relative mole percentage of mannose, glucuronic acid and xylose was 60.87:20.83:9.86).

### 2.2. Animal Experiment Design

C57BL/6J male mice (7-week-old, SPF-grade) were purchased from the Experimental Animal Center of Xi’an Jiaotong University (license number: SCXK 2018-001). The animal experiment protocol was approved by the Animal Ethics Committee of the Laboratory Animal Center of Northwest University (Approval Code: NWU-AWC-20200401M), and was carried out in accordance with the “Administrative Regulations on Laboratory Animals” of the National Science and Technology Commission of the People’s Republic of China. The mice were reared under standard conditions (temperature 25 ± 2 °C, relative humidity 40 ± 10%, light–dark cycle of 12/12 h, clean bedding, and water and food ad libitum). After one week of adaptation, the mice were randomly divided into 2 groups (*n* = 10): the mice on the normal diet (ND) group were fed with AIN93M standard feed (TROPHIC Animal Feed High-Tech Co., Ltd., Nantong, China) and distilled water, while the mice in the AAP group were given standard feed containing 200 mg/kg/day AAP (TROPHIC Animal Feed High-Tech Co., Ltd., Nantong, China) and distilled water. Both groups were fed for 12 weeks.

### 2.3. Collection of Blood and Faeces Samples

The formed faeces in the intestinal tract of mice were taken out and placed in a 2 mL sterile centrifuge tube, and stored at a −80 °C refrigerator for later use. Twenty-four hours after the last treatment, the mice were fasted without water overnight. Mouse blood was collected from the eyeballs and the plasma was centrifuged at 2500 r/min for 30 min at room temperature to obtain serum.

### 2.4. Fecal Bacteria 16S rRNA Sequencing Analysis

Genomic DNA of the samples were extracted by CTAB or SDS, and then the purity and concentrations were detected by agarose gel electrophoresis. An appropriate amount of DNA was taken and diluted to 1 ng/μL with sterile water. Using the diluted genomic DNA as a template, and according to the selection of sequencing region, PCR was performed using a specific primer with Barcode, Phusion^®^ High-Fidelity PCR Master Mix with GC Buffer from New England Biolabs, and high-performance high-fidelity enzyme, to ensure amplification efficiency and accuracy. Shanghai Baiqu Biomedical Technology Co., Ltd. (Shanghai, China) was commissioned to perform sequencing analysis on the V3-V4 region. The original data were spliced and filtered, followed by operational taxonomic unit (OTU) clustering and species classification analysis to obtain information on species richness and uniformity within the samples, as well as common and unique OTU information among different samples or groups. Principal coordinate analysis (PCoA), principal component analysis (PCA) and other dimensionality reduction methods were used to analyze the differences in community structure among the samples or groups. The Student’s *t* test, LEfSe and other statistical analysis methods were conducted to test the significance of species composition and community structure.

### 2.5. Extraction and Detection of Serum Metabolites

A 50 μL sample was transfer into an EP tube, to which was added 200 μL of an extract solution (methanol–acetonitrile 1:1 (*v*/*v*) containing an isotope-labeled internal standard mixture) (methanol: LC-MS grade, CNW Technologies; acetonitrile: LC-MS grade, CNW Technologies), followed by vortexing and mixing for 30 s. After ultrasonic treatment for 10 min in an ice water bath, the solution was left to stand at −40 °C for 1 h. Then, the samples were centrifuged at 4 °C at 12,000 rpm (centrifugal force 13,800× *g*; radius 8.6 cm) for 15 min. The supernatant was then collected into sample vials and subjected to computer testing. In addition, another equal amount of supernatant was taken from all samples and mixed into QC samples for on-machine testing.

The target compounds were separated by Waters ACQUITY UPLC BEH Amide (2.1 mm × 100 mm, 1.7 μm) liquid chromatography column using the Vanquish Ultra Performance Liquid Chromtography (UHPLC). Phase A was aqueous, containing 25 mmol/L ammonium acetate (LC-MS grade, Sigma-Aldrich) and 25 mmoL/L ammonia water (LC-MS grade, Fisher Chemical), Phase B was acetonitrile (LC-MS grade, CNW Technologies). The sample disc temperature was 4 °C and the injection volume was 2 μL. The Thermo Q Exactive HFX mass spectrometer was controlled by software (Xcalibur, Thermo) for primary and secondary mass spectrometry data acquisition. Detailed parameters were as follows: sheath gas flow rate, 30 Arb; aux gas flow rate, 25 Arb; capillary temperature, 350 °C; full ms resolution, 60,000; MS/MS resolution, 7500; Collision energy, 10/30/60 in NCE mode; spray voltage, 3.6 kV (positive) or −3.2 kV (negative).

The original data were converted into mzXML format by ProteoWizard software, and peak recognition, extraction, alignment and integration were processed by an R program package (kernel XCMS) independently written. Then, the converted data were matched with the BiotreeDB (V2.1) self-built secondary mass spectrometry database for material annotation. Metabolic markers were screened in combination with multivariate statistical analysis, and the algorithm score cutoff value was set to 0.3.

## 3. Results

### 3.1. The Effects of AAP on α Diversity of Intestinal Flora in Mice

As shown in Figure 1A, the dilution curves of the ND and AAP groups gradually smoothed out, indicating that the species in the intestinal tract were not enhanced by the increase in the number of sequenced samples, which in this study was sufficient to reflect the flora characteristics of the samples. The rank-abundance curve visually reflected the abundance and uniformity of the species. Dietary supplementation with AAP did not destroy the homeostasis of the intestinal flora species in the mice (Figure 1B); however, there were no statistically significant differences in the Chao1, ACE, Shannon, or Simpson indexes (Figure 1C–F).

### 3.2. The Effects of AAP on β Diversity of Intestinal Flora in Mice

β diversity mainly describes the changes in species composition on a spatial scale to evaluate the similarity between the groups; it can be analyzed by PCoA, which can visually show the similarities and differences in microorganisms in differently treated samples. As illustrated in Figure 2A, the intestinal microbial structure of the ND and AAP groups were similar, indicating that dietary supplementation of AAP did not markedly change the β diversity of the intestinal microbes. The box diagram for the group difference analysis among β diversity is shown in Figure 2B. The median, maximum and minimum sample values of the ND and AAP groups were different to some extent.

Anosim and MRPP analyses were used to determine whether there was a significant difference in community structure between ND and AAP groups. Anosim analysis is a non-parametric test that measures whether the difference between groups is significantly greater than the difference within a group to determine if the groups are meaningful. The significance test of the intergroup difference was performed based on the rank of the Bray–Curtis distance value. The analysis demonstrated that the R-value and *p*-value between the ND and AAP groups were 0.864 and 0.001, respectively, indicating that there was a significant difference between them. Furthermore, MRPP is a parametric test based on the Bray–Curtis distance. The smaller the observed delta value, the smaller the difference within the group; the larger the expected delta value, the greater the difference between the groups. A value greater than zero indicated that the intergroup difference was greater than that within each group. A value less than zero indicated that the difference within each group was greater than the difference between the groups. A significance value less than 0.05 indicates significant difference. The observed delta value, expected delta value and the significance value between the ND and AAP groups were 0.0717, 0.2683, 0.289 and 0.008, respectively, indicating that there was a significant difference in microbial community structure between the groups.

From Figure 2C, it can be seen that the relative abundance of intestinal flora in the mice had changed after AAP intake. At the phylum level, the microbial community structure of the mice feces samples was mainly composed of Bacteroidetes, Firmicutes, Actinobacteria, Campilobacter, Deferribacteres and Proteobacteria. Among these, Firmicutes and Bacteroidetes were the dominant flora, which was consistent with the results in the previous study. Therefore, the intestinal flora of the two groups of mice were similar. Compared with the ND group, the abundance of Bacteroidetes, Actinobacteria and Proteobacteria in the AAP group was significantly boosted, while that of Firmicutes was significantly decreased.

Through an intergroup Student’s *t* test, strains with significant difference (*p* < 0.05) were found as shown in Figure 2D. Dietary supplementation with AAP remarkedly increased expression of *Lactobacillus johnsonii*, *Weissella cibaria*, *Kosa Cosac Cowanii*, *Enterococcus faecalis*, *Bifidobacterium animalis* and *Bacteroides uniformis*, while significantly reducing the abundance of *Firmicutes bacterium M10-2*.

### 3.3. Screening of Mouse Serum Biomarkers after AAP Consumption 

A total of 239 substances (in negative ion mode) and 368 substances (in positive ion mode) were detected by mass spectrometry. The screening strategy of the serum biomarkers was as follows: multivariate statistical analysis was used to analyze the serum metabolic material spectrum data of the ND and AAP group in both modes, and the supervised analysis method––partial least squares discriminant analysis (OPLS-DA) ––was used to find, screen and identify the differential metabolites.

The OPLS-DA analysis of the two sets of data are shown in Figure 3A,B. The abscissa t(1)P represents represented the predicted principal component score of the first principal component, showing the differences between the groups, and the ordinate t(1)O represents the orthogonal principal component score, which shows the differences within each sample group. Each dot represents a sample, and the shape and color of the dots represent different experimental groups. From the results of the OPLS-DA score chart, the serum samples of the ND and AAP group mice were significantly different, and all samples were within the 95% confidence interval (Hotelling’s T-squared ellipse).

In addition, a replacement test was used to verify whether the model was over-fitted. As demonstrated in Figure 3C,D, the abscissa indicates the correlation between the random sample and the original sample, and the ordinate indicates the values of R^2^Y and Q^2^. In positive ion mode, R^2^Y(cum) and Q^2^(cum) of ND vs AAP were 0.96 and −0.14, while in negative ion mode R^2^Y(cum) and Q^2^(cum) were 0.99 and −0.11. The slope of R^2^Y was greater than 0, and the intercept of Q^2^ was less than 0.05, indicating that the model had good predictability and could be used for the next analysis without overfitting.

The results of screening for differential metabolites are visualized in the form of volcanic diagrams, as illustrated in Figure 3E,F. Each point in the diagram represents a metabolite. The abscissa represents multiple changes within a group in comparison to between groups (the logarithm with 2 as the bottom), and the ordinate represents the *p*-value of the Student’s *t* test (the negative number of the logarithm with 10 as the bottom). The scatter point size represents the variable importance projection (VIP) value of the OPLS-DA model. The larger the scatter point, the larger the value. The scatter colors represent the final screening results, with significantly up-regulated metabolites in red, significantly down-regulated metabolites in blue, and non-significantly different metabolites in grey. Substances that met the following requirements were identified as differential metabolites: (1) VIP value > 1; (2) significant difference *p* < 0.05; (3) exclusion of heterologous metabolites according to the references. The results showed that a total of 51 differential metabolites were screened out as potential biomarkers in the mice serum after the AAP dietary supplement. Of these, 1–20 were in positive ion mode and 21–54 were in negative ion mode. L-lysine, citrulline and indophenol sulfate were detected in both modes. Details of the selected biomarkers are shown in Table 1.

### 3.4. Changes in Serum Biomarkers in Mice after Auricularia auricula Polysaccharides Intake

The Euclidean distance matrix was calculated based on the quantitative values of the differential metabolites; the differential metabolites were clustered using the complete linkage method and displayed in the thermal diagram. In Figure 4A, the abscissa represents different experimental groups, and the ordinate represents the comparative differential metabolites of the group (consistent with Table 1). The color patches at different positions represent the relative expression levels of the metabolites at the corresponding positions. Red indicates that the substance content was highly expressed, and blue indicates that it was low.

The results of the metabolic pathway analysis are presented as bubble charts (Figure 4B) where each bubble represents a metabolic pathway. The abscissa of the bubble and the size of the bubble represents the size of the pathway influence factor in topology analysis. The larger the size, the larger the influence. The vertical coordinate where the bubble was located and the color of the bubble indicated the *p* value (negative natural log, i.e., −ln (*p*)) of the enrichment analysis. The darker the color, the smaller the *p* value, and the more significant the degree of enrichment. As shown in Figure 4 B, the differential genes caused by the AAP dietary supplement were mainly enriched in the arginine biosynthesis pathway, which had the highest correlation; this was followed by the arginine and proline metabolism; glycine, serine and threonine metabolism; glycerophospholipid metabolism; and sphingolipid metabolism pathways.

### 3.5. Correlation Analysis of Intestinal Flora and Metabolites in Mice after Intake of Auricularia auricula Polysaccharides

To further investigate the correlation between the intestinal flora of mice and the metabolites of the body after intake of AAP, the content of the metabolites with an abundance of significantly different flora (*p* ≤ 0.01) were analyzed by thermography. The results are illustrated in Figure 5. There were obvious positive correlations between *Weissella cibaria* and the following metabolites: capric acid, polyoxyethylene (600) monoricinoleate, guanidoacetic acid, 12-oxo-2,3-dinor-10,15-phytodienoic acid, 2-acetyl-3,6-dimethylpyrazine, 3-methoxy-4-hydroxyphenylethyleneglycol sulfate, D-malic acid, indoxyl, indoxyl sulfate, isokobusone, phenylglyoxylic acid. There were also markedly negative correlations with (-)-matairesinol, D-ribose, eicosapentaenoic acid, sedoheptulose, solasodine. *Lactobacillus johnsonii* and 3-methoxy-4-hydroxyphenylethyleneglycol sulfate, xanthosine, 12-oxo-2,3-dinor-10,15-phytodienoic acid, capric acid, D-malic acid, guanidoacetic acid, hippuric acid, isokobusone, L-lysine, phenylglyoxylic acid, polyoxyethylene (600) monoricinoleate, and trans-aconitic acid correlated positively, while (-)-matairesinol, D-ribose, eicosapentaenoic acid, and sedoheptulose correlated negatively; it was found that there were obvious positive correlations between *Kosakonia cowanii* and polyoxyethylene (600) monoricinoleate, 12-oxo-2,3-dinor-10,15-phytodienoic acid, 2-acetyl-3,6-dimethylpyrazine, 3-methoxy-4-hydroxyphenylethyleneglycol sulfate, capric acid, D-malic acid, hippuric acid, indoxyl, isokobusone, L-lysine, and xanthosine, while there were negative correlations with (-)-matairesinol, and eicosapentaenoic acid. *Firmicutes bacterium M10-2* and metabolites such as (-)-matairesinol, D-ribose, eicosapentaenoic acid, sedoheptulose correlated positively, but negatively with isokobusone, 2-acetyl-3,6-dimethylpyrazine, benzoic acid, capric acid, guanidoacetic acid, hippuric acid, indoxyl, indoxyl sulfate, isokobusone, L-lysine, phenylglyoxylic acid, polyoxyethylene (600) monoricinoleate, and xanthosine. There were obvious positive correlations between *Enterococcus faecalis* and the following metabolites: (R)-dihydromaleimide, 12-oxo-2,3-dinor-10,15-phytodienoic acid, 2-acetyl-3,6-dimethylpyrazine, benzoic acid, D-malic acid, indoxyl, indoxyl sulfate, L-lysine, phenylglyoxylic acid, polyoxyethylene (600) monoricinoleate, and trans-aconitic acid. There were also markedly negative correlations with D-ribose, eicosapentaenoic acid, sedoheptulose, and solasodine. *Bifidobacterium animalis* and the following metabolites correlated positively: D-malic acid, trans-aconitic acid, xanthosine, (R)-dihydromaleimide, 12-oxo-2,3-dinor-10,15-phytodienoic acid, 2-acetyl-3,6-dimethylpyrazine, 3-methoxy-4-hydroxyphenylethyleneglycol sulfate, hippuric acid, indoxyl, indoxyl sulfate, isokobusone, L-lysine, phenylglyoxylic acid, and polyoxyethylene (600) monoricinoleate. Negative correlations were found with (-)-matairesinol, D-ribose, eicosapentaenoic acid, sedoheptulose, solasodine; it was found that there were obvious positive correlations between *Bacteroides uniformis* and indoxyl sulfate, (R)-dihydromaleimide, 12-oxo-2,3-dinor-10,15-phytodienoic acid, 2-acetyl-3,6-dimethylpyrazine, benzoic acid, D-malic acid, indoxyl, isokobusone, L-lysine, phenylglyoxylic acid, polyoxyethylene (600) monoricinoleate, trans-aconitic acid, and xanthosine. (-)-matairesinol and eicosapentaenoic acid correlated negatively.

## 4. Discussion

The United Nations has proposed “one meat, one vegetable and one mushroom” as a reasonable dietary structure for the 21st century. Polysaccharides are formed by the polymerization of more than 10 monosaccharides through glycosidic bonds that have high safety and low toxicity. Polysaccharides from edible and pharmaceutical bacteria are considered to be “biological response regulators” [16]. At present, more than 100 kinds of edible and medicinal fungal polysaccharides have been found: antioxidant, antidiabetic, hypolipidemic, immunomodulatory, antitumor and liver protective [17], among which *Lentinus edodes*, *Ganoderma lucidum*, *Cordyceps sinensis* and *Tremella fuciformis* have been widely used in medicine, life sciences and food fields [18]. Due to the lack of enzymes, most of the edible and medicinal fungal polysaccharides cannot be directly digested and absorbed by the body, while CAZymes encoded by intestinal microflora can convert oligosaccharides and polysaccharides into monosaccharides to generate SCFAs and other metabolites that are easy absorbed [19]. In addition, different types of polysaccharides can increase beneficial intestinal microorganisms and reduce harmful intestinal microorganisms [13]. HAO et al. found that after 28 days of administering *Flammulina velutipes* polysaccharides, the abundance of *Firmicutes* in the cecum of C57BL/6J mice increased, while the abundance of *Bacteroides* decreased, thus increasing the ratio of *Firmicutes* to *Bacteroides* [20]. KHAN et al. studied the effects of *ganoderma lucidum* polysaccharides and *poria cocos* polysaccharides on the intestinal flora of C57BL/6J mice; they found that two edible and medicinal fungal polysaccharides increased production of SCFAs and lactic acid and the abundance of probiotics resistant to obesity such as *Bifidobacterium*, *Eubacterium rectal*, *Lactobacillus* and *Lactococcus.* In addition, the proportion of pathogenic bacteria was reduced; thus playing a prebiotic role [21]. The results of this study demonstrated that dietary supplementation of AAP changed the composition of the intestinal flora in C57BL/6J male mice and significantly up-regulated *Lactobacillus johnsonii*, *Weissella cibaria*, *Kosakonia cowanii*, *Enterococcus faecalis*, *Bifidobacterium animalis* and *Bacteroides uniformis*, while down-regulating the relative abundance of *Firmicutes bacterium M10-2*.

There are hundreds of millions of bacterial communities in the intestines of mammals, which are interdependent of their hosts and participate in various physiological activities: metabolism, immunity, and regulation of endocrine and nervous system functions [22]. Emerging evidence suggests that low-grade inflammation is a marker of metabolic disorders such as obesity, type 2 diabetes, and non-alcoholic fatty liver disease [23,24]; these diseases are characterized by changes in the intestinal microbiota and its metabolites, which migrate from the intestinal tract through the damaged intestinal barrier, affecting metabolic organs such as the liver and adipose tissue [25]. Other studies found that obesity, the propensity to gain weight, dyslipidemia, insulin resistance and low-grade inflammation were more prevalent in subjects exhibiting low gut bacterial richness [26]. Additionally, it was suggested that certain “proinflammatory” bacterial strains such as *Ruminococcus gnavus* or *Bacteroides* species, might dominate, while “anti-inflammatory”’ strains, such as *Faecalibacterium prausnitzii*, are less prevalent [27]. Lactic acid bacteria (LAB) are best known for imparting beneficial health effects. Numerous studies showed that lactic acid bacteria resist oxidation, regulate immune function, reduce cholesterol, promote digestion and prevent cancer; thus the inclusion in the diet could be positive [28]. By definition, LAB produces large amounts of lactic acid using carbohydrates. *Lactococcus*, *Enterococcus*, *Pediococcus*, *Leuconostoc*, *Lactobacillus*, *Bifidobacterium*, *Streptococcus*, *Vagococcus*, *Tetragenococcus* and *Weissella* are some lactic acid bacterial genera that have been well studied for their health benefits [29]. As a probiotic, *Lactobacillus johnsonii* plays an important role in maintaining the balance of intestinal flora and regulating the immune system. Studies have found that it promotes the growth and development of animals, reduces inflammatory reactions, prevents diarrhea, increases the number of beneficial bacteria in the intestine, and regulates the balance of intestinal microflora [30]. After colonization by *Lactobacillus yoelii*, the number of CD^4+^ and CD^8+^ cells in the small intestine and spleen were the highest. Although less effective than fecal bacterial transplantation, colonization with *Escherichia coli* and *Lactobacillus johnsonii* both increased the proportion of regulatory T cells (Tregs) and activated DC and completely restored the intestinal memory/effector T cell population on Day 28. Interestingly, only *L. johnsonii* recolonization maintained colonic IL-10 production [31]. Other studies demonstrated that *L. johnsonii* increases the level of reduced glutathione in the blood, thus improving the mitochondrial morphology and function of the liver by reducing liver lipids and improving systemic glucose metabolism [32]. *Weissella cibaria* is the dominant heterotypic lactic acid fermentation bacterium found in the early stage of the natural fermentation of pickled cabbage. Due to the complexity of fermentation and the diversity of products, it can produce more flavor substances and give pickled cabbage a better flavor [33]. Shashank et al. isolated four new Weiss strains from fermented dosa batter and human infant fecal sample and found that all four tolerated gastric fluid (pH = 3.0) and bile salt, moderate cell surface hydrophobicity, reduced cholesterol, and adhered to CaCo2 cells and gastric mucin. The cell experiment proved that all four strains inhibited production of nitric oxide and IL-6 by a mouse macrophage induced by a lipopolysaccharide and inhibited the production of IL-8 in human epithelial cells [34]. In this study, it was found that dietary AAP supplementation can significantly up-regulate the abundance of *L. johnsonii* and *Weissella cibaria*, but its health promotion effect based on the regulation of intestinal flora needs to be further explored.

Different polysaccharides promote the growth of different intestinal flora, which is related to the structure of the polysaccharides. The relative molecular mass is an important factor affecting the biological activity of polysaccharides. You et al. compared the structural information, digestive behavior and effects on the intestinal microflora of three brown algal polysaccharides (alginate, laminarin, fucoidan), and found that the low molecular-weight polysaccharides had a greater effect on the intestinal microflora regulation [35]. Deng et al. explored the physicochemical properties, hypoglycemic effects, and mechanism analysis of low, medium and high molecular weight konjac glucomannan (KGM) and found that the medium molecular weight altered the ratio of *Firmicutes* to *Bacteroides* and reduced the abundance of *Roma Boots* and *Klebsiella*. The *Bacteroides* population increased by 6.14, 24.2, 18.38, and 12.28% in the KGM-H (1129.5 kDa), KGM-M1 (757.1 kDa), KGM-M2 (252.7 kDa), and KGM-L (87.3 kDa) groups, respectively [36]. Furthermore, the beneficial effects of polysaccharides are closely related to the monosaccharide types. β-glucan extracted from oat increased the number of *Bifidobacterium* and *Lactobacillus* in intestinal microflora [37]. *Bacteroides* effectively degraded fructan [38]; and xylan was metabolized by *Prevotella bryantii* [39]. In this study, the polysaccharides of *Auricularia auricula* from the Qinba Mountains prepared by hot water extraction was mainly composed of mannose (the relative molar percentage of mannose, glucuronic acid and xylose is 60.87: 20.83:9.86) [4], and its regulating effect on intestinal flora may be closely related to mannose. Studies found that mannose is not only an important monosaccharide for protein glycosylation in mammals but also an inefficient source of cell energy. Supplementing a certain amount of mannose could increase the ratio of *Bacteroides* to *Firmicutes* in the intestinal microflora of mice fed a high-fat diet. Functional transcriptomic analysis of the mice cecal microflora revealed that mannose induced coherent changes in microbial energy metabolism, and it was speculated that its effect came from reducing the energy produced by intestinal microbiota that metabolize complex carbohydrates and reduce energy intake [40]. Furthermore, Li et al. analyzed the effects of four galactoses (porphyrin, agarose, carrageenan, and arabinogalactan) with significant differences in molecular weight, polymerization glycosidic bonds, esterification, branching, and monosaccharide composition on intestinal microflora and SCFAs production. The results showed that all four galactoses can be used by intestinal *bacteroides*. In addition, the porphyran can be used by *Lactobacillus* and *Bifidobacterium*, while the arabinogalactan can be used by *Lactobacillus*, *Bifidobacterium* and *Roseburia* [41]. Although a large number of studies confirmed that the structure of polysaccharides affects their regulatory function of various intestinal microflora, how the intestinal microorganisms use different types of polysaccharides according to their structural characteristics remains to be studied further. Due to the lack of research into the primary and secondary polysaccharide structures, the different origins of edible and medicinal fungi, and the different polysaccharides extraction methods, the corresponding quality control indexes are lacking, which affects the repeatability of experiments and adds difficulty to the development of medicinal and nutrition products.

The body depends on the glycolysis of intestinal flora to absorb nutrients from the food that cannot be digested by its own enzymes, and to obtain energy. At the same time, intestinal flora regulate the normal physiological functions of the body and the occurrence and development of diseases [42]. Based on the metabolomics platform, this study carried out a systematic analysis on the serum of the endogenous metabolite spectrum of mice fed AAP from the Qinba Mountains, and analyzed the correlation between significant changes in intestinal flora and metabolic differences after AAP intake. The results demonstrated that AAP had a significant regulatory effect on the serum metabolite spectrum, and the metabolites had a marked correlation with *Lactobacillus johnsonii*, *Weissella cibaria*, *Kosakonia cowanii*, *Enterococcus faecalis*, *Bifidobacterium animalis*, *Bacteroides uniformis* and *Firmicutes bacterium M10-2*. After AAP intake, the differential genes were mainly enriched in the arginine biosynthesis pathway, where the correlation was the highest. L-arginine is a basic semi-essential amino acid. When the animal body is injured (especially the liver) or stressed, a large amount of arginase is released into the blood to accelerate the consumption of arginine, which also causes a lack of arginine [43,44]. There are four main pathways of arginine metabolism in mammals. One is the formation of ornithine and urea with ammonia catalyzed by arginase, also known as the urea cycle, which is also an important means of ammonia metabolism [45]. Urea produced by metabolism is excreted through the urine, and the main site of urea circulation is the liver. Ornithine can be decarboxylated by ornithine decarboxylase to generate putrescine, which further reacts with S-adenosylmethionine to generate spermidine and spermine. Putrescine, spermidine and spermine are collectively referred to as polyamines, which are substances that can regulate the synthesis of body DNA and protein, thus promoting cell proliferation and differentiation. The second pathway is the formation of agmatine under the catalysis of arginine decarboxylase, which has been proven to be a new neurotransmitter that can affect the release of hormones and promote the proliferation of lymphocytes and thymocytes. Agmatine can produce putrescine catalyzed by the agmatine enzyme, which is also another way that polyamine is produced in the body [46]. Third, guanidinoacetic acid is generated by the catalysis of arginine: glycine transferase and guanidinoacetic acid are methylated under the catalysis of guanidinoacetic acid-N-methyltransferase to form creatine, the main synthesis sites of which are the kidneys and pancreas. During blood circulation, creatine can be absorbed by skeletal muscle and neurons and further phosphorylated to form phosphocreatine, which is mainly involved in energy metabolism in the skeletal muscle [47]. In addition, creatine can also be irreversibly converted into dehydrated creatinine through a non-enzymatic pathway. The last metabolic pathway of arginine is the production of nitric oxide (NO) and citrulline catalyzed by nitric oxide synthase (NOS). NO is a crucial regulator in mammalian tissue cells, and citrulline can be used as a substrate to regenerate arginine [48].

Further studies are needed to explore the relationships among the “active components–intestinal flora–body metabolism” of *Auricularia auricula* and its effects on gut health and related metabolic diseases to develop and use edible and medicinal fungi resources.

## 5. Conclusions

This study showed that non-digestible AAP promoted the growth of *Lactobacillus johnsonii*, *Weissella cibaria*, *Kosakonia cowanii*, *Enterococcus faecalis*, *Bifidobacterium animalis* and *Bacteroides uniformis* in normal mice intestines. After AAP dietary supplementation, 51 differential metabolites were found, which were mainly enriched in arginine biosynthesis pathways, followed by arginine and proline metabolism; glycine, serine and threonine metabolism; glycerophospholipid metabolism; and the sphingolipid metabolism pathway. Furthermore, it was found that there is a close correlation between differential bacteria and differential metabolites, which may contribute to the bioactivity of AAP; these results laid the theoretical basis for the further development and use of *Auricularia auricula* resources from the Qinba Mountains and nutritional foods with AAP as the main functional component.

## Figures and Tables

**Figure 1 foods-11-02700-f001:**
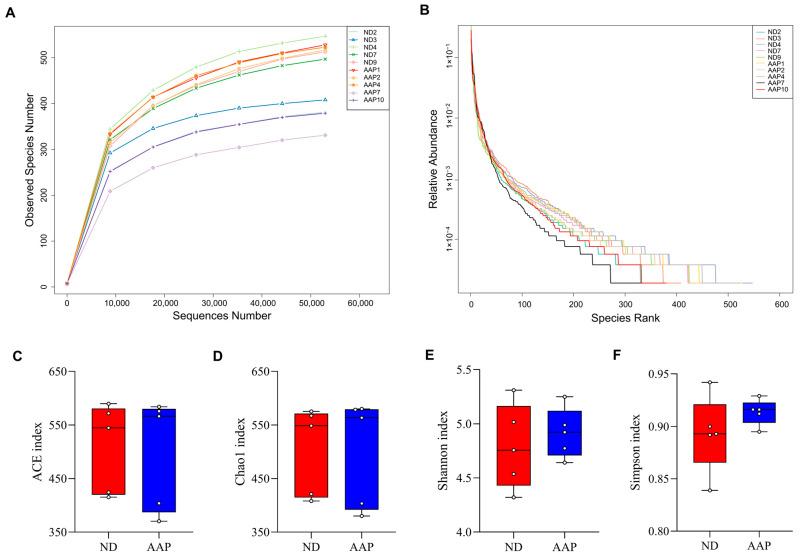
The effects of *Auricularia auricula* polysaccharides on α diversity of intestinal flora in mice. (**A**) Dilution curve; (**B**) Rank Abundance curve; (**C**) Group difference box diagram of the ACE index; (**D**) Chao 1 index; (**E**) Shannon index; (**F**) Simpson index. ND = normal diets; AAP = *Auricularia auricula* polysaccharides. Data presented as mean ± SD, *n* = 5.

**Figure 2 foods-11-02700-f002:**
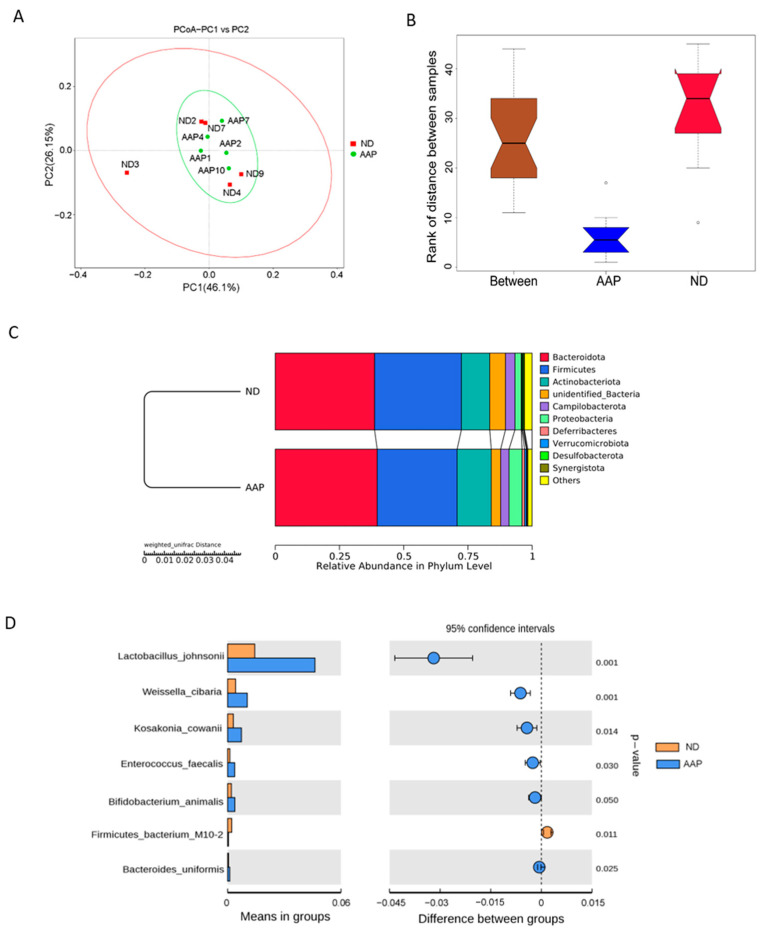
The effects of *Auricularia auricula* polysaccharides on β diversity of intestinal flora in mice. (**A**) Principle component analysis (PCoA) based on weighted unifrac distance; (**B**) Anosim difference analysis between groups; (**C**) Unweighted pair group method with arithmetic mean (UPGMA) clustering tree based on weighted unifrac distance; (**D**) Group species difference analysis chart of Student’s *t* test at the species level. ND = normal diets; AAP = *Auricularia auricula* polysaccharides. Data presented as mean ± SD, *n* = 5.

**Figure 3 foods-11-02700-f003:**
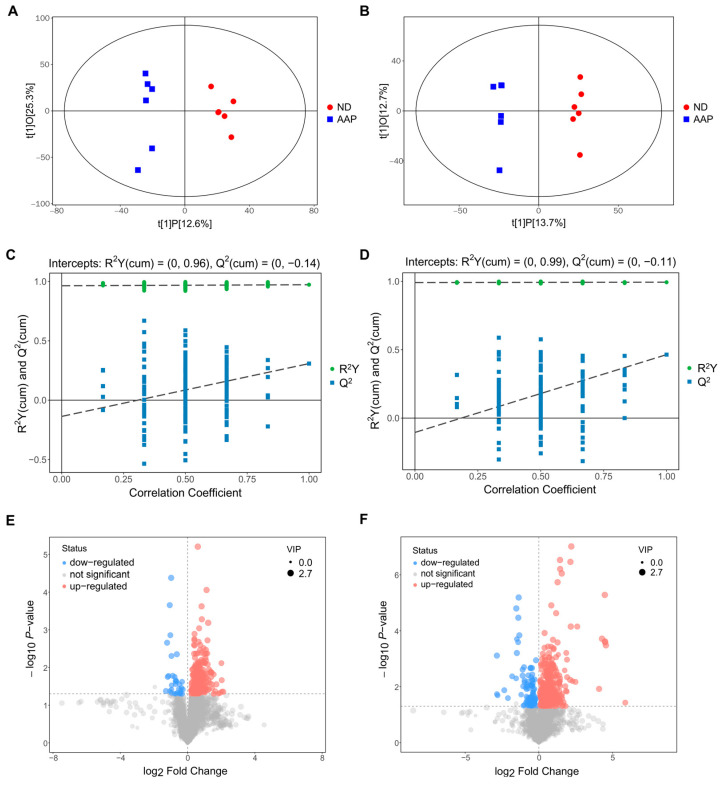
Biomarkers identification after Auricularia auricula polysaccharides consumption by serum metabolomics. OPLS-DA score plot in (**A**) positive ion mode; (**B**) negative ion mode; OPLS-DA replacement test (200 times) in (**C**) positive ion mode; (**D**) negative ion mode; The slope of R^2^ is greater than 0, and the intercept of Q^2^ on the Y axis is less than 0.05, indicating that the model is effective; Volcano map of differential metabolite screening in (**E**) positive ion mode; (**F**) negative ion mode. ND = normal diets; AAP = *Auricularia auricula* polysaccharides. Data presented as mean ± SD, *n* = 6.

**Figure 4 foods-11-02700-f004:**
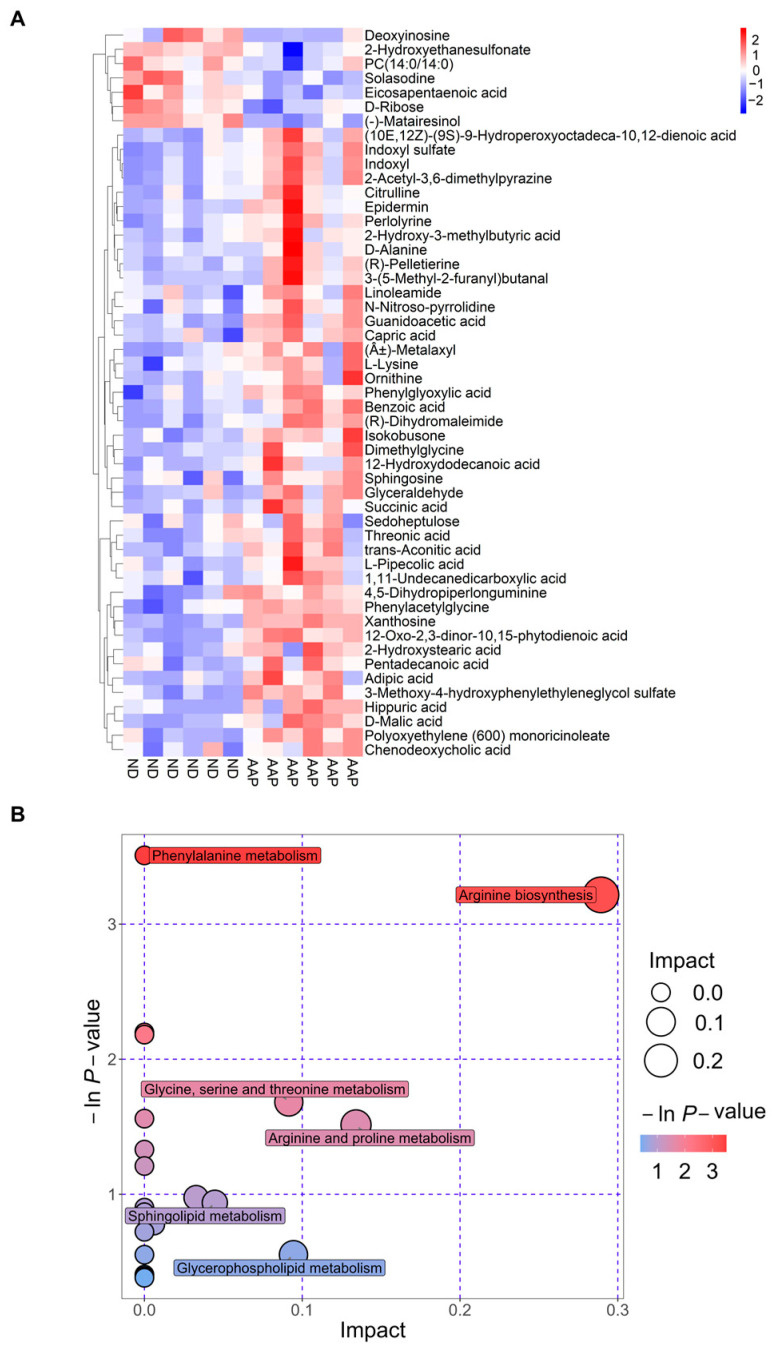
Metabolic pathway analysis. (**A**) Thermogram of serum biomarker changes; (**B**) Bubble chart. ND = normal diets; AAP = *Auricularia auricula* polysaccharides.

**Figure 5 foods-11-02700-f005:**
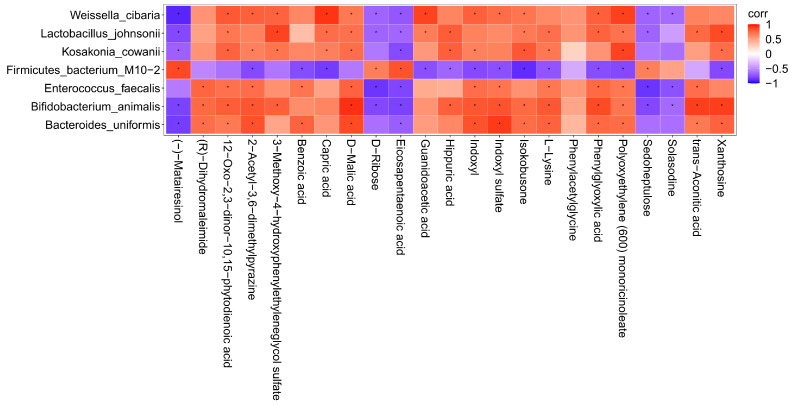
Correlation analysis between characteristic bacteria and metabolites. corr = correlation. Red represents positive correlation and blue represents negative correlation. The deeper the color, the greater the correlation. * indicates *p* < 0.05.

**Table 1 foods-11-02700-t001:** Identification of mice serum metabolites after administering *Auricularia auricula* polysaccharides.

No.	Metabolites	Sub Class	VIP Value	*p* Value	Experimental *m*/*z*	Molecular Formula	HMDB ID	KEGG ID	AAP vs. ND
1	Polyoxyethylene (600) monoricinoleate	Fatty alcohols	2.286	0.001	341.3039	C_21_H_40_O_3_	HMDB0032476	---	↑
2	Solasodine	Steroidal alkaloids	2.286	0.004	414.3364	C_27_H_43_NO_2_	HMDB0035282	C10822	↓
3	Guanidoacetic acid	Amino acids, peptides, and analogues	2.240	0.001	118.0612	C_3_H_7_N_3_O_2_	HMDB0000128	C00581	↑
4	Epidermin	Carbohydrates and carbohydrate conjugates	2.194	0.020	262.1273	C_11_H_19_NO_6_	HMDB0034777		↑
5	(R)-Dihydromaleimide	Pyrrolines (Class)	2.094	0.005	100.0397	C_4_H_5_NO_2_	HMDB0030276	---	↑
6	(R)-Pelletierine	Piperidines (Class)	2.053	0.041	142.1224	C_8_H_15_NO	HMDB0030324		↑
7	Indoxyl sulfate	Arylsulfates	2.031	0.006	214.0163	C_8_H_7_NO_4_S	HMDB0000682	---	↑
8	3-(5-Methyl-2-furanyl)butanal	Heteroaromatic compounds (Class)	2.016	0.047	153.0907	C_9_H_12_O_2_	HMDB0036169	---	↑
9	2-Acetyl-3,6-dimethylpyrazine	Carbonyl compounds	1.981	0.008	151.0861	C_8_H_10_N_2_O	HMDB0039998	---	↑
10	Indoxyl	Hydroxyindoles	1.970	0.010	134.0598	C_8_H_7_NO	HMDB0004094	C05658	↑
11	4,5-Dihydropiperlonguminine	Benzodioxoles (Class)	1.957	0.025	276.1587	C_16_H_21_NO_3_	HMDB0041488	---	↑
12	Perlolyrine	Harmala alkaloidal (Class)	1.915	0.020	265.0966	C_16_H_12_N_2_O_2_	HMDB0030327	C09231	↑
13	N-Nitroso-pyrrolidine	Pyrrolidines (Class)	1.741	0.037	101.0712	C_4_H_8_N_2_O	HMDB0031642	C19285	↑
14	Citrulline	Amino acids, peptides, and analogues	1.734	0.043	176.1025	C_6_H_13_N_3_O_3_	HMDB0000904	C00327	↑
15	L-Lysine	Amino acids, peptides, and analogues	1.667	0.040	147.1125	C_6_H_14_N_2_O_2_	HMDB0000182	C00047	↑
16	(Â±)-Metalaxyl	Amino acids, peptides, and analogues	1.648	0.037	280.1532	C_15_H_21_NO_4_	HMDB0031802	C10947	↑
17	PC(14:0/14:0)	Glycerophosphocholines	1.625	0.049	678.5077	C_36_H_72_NO_8_P	HMDB0007866	C00157	↓
18	Ornithine	Amino acids, peptides, and analogues	1.614	0.047	133.0969	C_5_H_12_N_2_O_2_	HMDB0000214	C00077	↑
19	Linoleamide	Fatty amides	1.361	0.050	280.2626	C_18_H_33_NO	HMDB0062656	---	↑
20	Sphingosine	Amines	1.358	0.036	300.2883	C_18_H_37_NO_2_	HMDB0000252	C00319	↑
21	Xanthosine	Purine nucleosides (Class)	2.569	0.000	283.0687	C_10_H_12_N_4_O_6_	HMDB0000299	C01762	↑
22	3-Methoxy-4-hydroxyphenylethyleneglycol sulfate	Arylsulfates	2.301	0.000	263.0233	C_9_H_12_O_7_S	HMDB0000559	---	↑
23	(-)-Matairesinol	Tetrahydrofuran lignans	2.274	0.000	357.1350	C_20_H_22_O_6_	HMDB0035698	C10682	↓
24	Dimethylglycine	Amino acids, peptides, and analogues	2.183	0.020	102.0551	C_4_H_9_NO_2_	HMDB0000092	C01026	↑
25	Isokobusone	Alcohols and polyols	2.156	0.003	221.1548	C_14_H_22_O_2_	HMDB0036791	---	↑
26	Indoxyl sulfate	Arylsulfates	2.146	0.002	212.0020	C_8_H_7_NO_4_S	HMDB0000682	---	↑
27	12-Oxo-2,3-dinor-10,15-phytodienoic acid	Eicosanoids	2.134	0.000	263.1658	C_16_H_24_O_3_	HMDB0032090	---	↑
28	Eicosapentaenoic acid	Fatty acids and conjugates	2.078	0.006	301.2173	C_20_H_30_O_2_	HMDB0001999	C06428	↓
29	D-Malic acid	Beta hydroxy acids and derivatives	2.065	0.004	133.0134	C_4_H_6_O_5_	HMDB0031518	C00497	↑
30	Phenylacetylglycine	Amino acids, peptides, and analogues	2.046	0.005	192.0662	C_10_H_11_NO_3_	HMDB0000821	C05598	↑
31	trans-Aconitic acid	Tricarboxylic acids and derivatives	2.019	0.007	173.0085	C_6_H_6_O_6_	HMDB0000958	C02341	↑
32	Adipic acid	Lipids and lipid-like molecules	2.005	0.013	145.0497	C_6_H_10_O_4_	HMDB0000448	C06104	↑
33	Benzoic acid	Benzoic acids and derivatives	1.991	0.006	121.0285	C_7_H_6_O_2_	HMDB0001870	C00539	↑
34	D-Ribose	Carbohydrates and carbohydrate conjugates	1.988	0.008	149.0447	C_5_H_10_O_5_	HMDB0000283	C00121	↓
35	L-Lysine	Amino acids, peptides, and analogues	1.951	0.007	145.0974	C_6_H_14_N_2_O_2_	HMDB0000182	C00047	↑
36	Capric acid	Fatty acids and conjugates	1.941	0.007	171.1384	C_10_H_20_O_2_	HMDB0000511	C01571	↑
37	(10E,12Z)-(9S)-9-Hydroperoxyoctadeca-10,12-dienoic acid	Lineolic acids and derivatives	1.930	0.016	311.2235	C_18_H_32_O_4_	HMDB0062434	C14827	↑
38	Sedoheptulose	Carbohydrates and carbohydrate conjugates	1.921	0.009	209.0662	C_7_H_14_O_7_	HMDB0003219	C08355	↓
39	D-Alanine	Amino acids, peptides, and analogues	1.918	0.015	88.0396	C_3_H_7_NO_2_	HMDB0001310	C00133	↑
40	Succinic acid	Dicarboxylic acids and derivatives	1.917	0.020	117.0184	C_4_H_6_O_4_	HMDB0000254	C00042	↑
41	12-Hydroxydodecanoic acid	Medium-chain hydroxy acids and derivatives	1.876	0.023	215.1653	C_12_H_24_O_3_	HMDB0002059	C08317	↑
42	Citrulline	Amino acids, peptides, and analogues	1.800	0.034	174.0877	C_6_H_13_N_3_O_3_	HMDB0000904	C00327	↑
43	2-Hydroxy-3-methylbutyric acid	Fatty acids and conjugates	1.775	0.033	117.0547	C_5_H_10_O_3_	HMDB0000407	---	↑
44	L-Pipecolic acid	Amino acids, peptides, and analogues	1.731	0.029	128.0707	C_6_H_11_NO_2_	HMDB0000716	C00408	↑
45	Glyceraldehyde	Carbohydrates and carbohydrate conjugates	1.694	0.039	89.0233	C_3_H_6_O_3_	HMDB0001051	C02154	↑
46	Deoxyinosine	Purine 2′-deoxyribonucleosides	1.682	0.020	251.0791	C_10_H_12_N_4_O_4_	HMDB0000071	C05512	↓
47	Hippuric acid	Benzoic acids and derivatives	1.656	0.006	178.0505	C_9_H_9_NO_3_	HMDB0000714	C01586	↑
48	Pentadecanoic acid	Fatty acids and conjugates	1.655	0.041	241.2176	C_15_H_30_O_2_	HMDB0000826	C16537	↑
49	Phenylglyoxylic acid	Benzoyl derivatives	1.644	0.005	149.0236	C_8_H_6_O_3_	HMDB0001587	C02137	↑
50	Chenodeoxycholic acid	Lipids and lipid-like molecules	1.622	0.044	391.2871	C_24_H_40_O_4_	HMDB0000518	C02528	↑
51	Threonic acid	Carbohydrates and carbohydrate conjugates	1.576	0.046	135.0291	C_4_H_8_O_5_	HMDB0000943	C01620	↑
52	1,11-Undecanedicarboxylic acid	Fatty acids and conjugates	1.463	0.049	243.1605	C_13_H_24_O_4_	HMDB0002327	---	↑
53	2-Hydroxystearic acid	Fatty acids and conjugates	1.452	0.039	299.2592	C_18_H_36_O_3_	HMDB0062549	C03045	↑
54	2-Hydroxyethanesulfonate	Organosulfonic acids and derivatives	1.303	0.029	124.9904	C_2_H_6_O_4_S	HMDB0003903	C05123	↓

ND = normal diets; AAP = *Auricularia auricula* polysaccharides. VIP = Variable importance projection; ↑ = up-regulated; ↓ =down-regulated.

## Data Availability

The data presented in the manuscript is available on request from the corresponding author.

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
