# Peer review of "Effects of Auricularia auricula Polysaccharides on Gut Microbiota and Metabolic Phenotype in Mice"

_foods, 2022, doi:10.3390/foods11172700_

Round 1

Reviewer 1 Report

This is a good MS on a very interesting topic. The authors are invited to carry out some edits as below:

1. lines 33 and 34, please delete "obviously",

2. lines 134-137, the verbs should be given in passive voice, i.e. 30grams were weighed etc,

3. line 171 : spelling of "faeces" (not feces) and in other places where this word is used,

4. lines 200-202 : verbs need to be in passive voice,

5. figure 1 is of very poor quality! in A and B parts of the figure, the legend is tiny and it can not be read, also all abbreviations used should be given in full in the figure legend,

6. the same applies for figure 2, in A part of the figure, I can't read the info for each spot,

7. figure 3: the fonts used are very small and not visible to the reader,

8. in table 1: what does VIP mean? Also, please give in full the word "exptl",

9. figure 4: fonts are again very small,

10. figure 5: please give in the legend the physical meaning of red and blue colours,

11. in discussion, after line 583, the authors are invited to include one more paragraph highlighting the main outcomes of their work in relation to composition changes of gut microflora and relate these changes to possible anti-inflammatory effects.

12. The role of inflammation is not mentioned at all in the MS. I would invite the authors to include some discussion on inflammation and chronic diseases. Useful references that the authors may wish to consider:  Nutrients | Free Full-Text | Cholesterol versus Inflammation as Cause of Chronic Diseases (mdpi.com)

and Nutrients | Free Full-Text | Inflammation, not Cholesterol, Is a Cause of Chronic Disease (mdpi.com)

I am happy to review the revised MS in due course.

Author Response

Dear Editor and Reviewers,

    Thank you for your helpful comments on our manuscript entitled “Effects of Auricularia auricula polysaccharides on gut microbiota and metabolic phenotype in mice” (foods- 1854627) that helps improve the quality of the study. The manuscript has been polished by MDPI language editing services (https://www.mdpi.com/authors/english), and the revised portions are highlighted in blue in the revised manuscript. The main corrections in the manuscript and the responses to the reviewers’ comments are listed below point by point.

Reviewer 1:

This is a good MS on a very interesting topic. The authors are invited to carry out some edits as below:

Responses:

    Thank you for your encouraging comments. We have studied comments carefully and revised our manuscript seriously. The manuscript has been polished by MDPI language editing services (https://www.mdpi.com/authors/english), and the revised portions are highlighted in blue in the revised manuscript. We hope that the corrections will meet with approval.

Comments:

  1. lines 33 and 34, please delete "obviously",

Responses:

    Thanks for the reviewer’s kind remind. We have deleted "obviously" in the revised manuscript, please see in Page 2, Line 33.

Comments:

  1. lines 134-137, the verbs should be given in passive voice, i.e. 30grams were weighed etc,

Responses:

    Thanks for the reviewer’s kind remind. We have re-written the sentence in passive voice in the revised manuscript.

    Specifically, as follows: [Thirty grams of Auricularia auricula-judae powder, purchased from Shaanxi Tianmei Green Industry Co., Ltd., and grown in the Qinba Mountains of northwest China, were weighed, and extracted using 100 °C water (solid–liquid ratio 1:50) for 2 h.]

Comments:

  1. line 171 : spelling of "faeces" (not feces) and in other places where this word is used,

Responses:

    Thanks for the reviewer’s kind remind. We have replaced “feces” with "faeces" in the whole revised manuscript.

Comments:

  1. lines 200-202 : verbs need to be in passive voice,

Responses:

    Thanks for the reviewer’s kind remind. We have re-written the sentence in passive voice in the revised manuscript. The manuscript has been polished by MDPI language editing services, and the revised portions are highlighted in blue in the revised manuscript. We hope that the corrections will meet with approval.

Comments:

  1. figure 1 is of very poor quality! in A and B parts of the figure, the legend is tiny and it can not be read, also all abbreviations used should be given in full in the figure legend,

Responses:

    Thanks for the reviewer’s kind remind. In the revised manuscript, Figure 1 has been redrawn to enhance readability. Besides, we also uploaded separate high-definition figures in the system. We have added the full name of abbreviations in the figure legend. We hope that the corrections will meet with approval.

Comments:

  1. the same applies for figure 2, in A part of the figure, I can't read the info for each spot,

Responses:

    Thanks for the reviewer’s kind remind. In the revised manuscript, Figure 2 has been redrawn to enhance readability. Besides, we also uploaded separate high-definition figures in the system. We hope that the corrections will meet with approval.

Comments:

  1. figure 3: the fonts used are very small and not visible to the reader,

Responses:

    Thanks for the reviewer’s kind remind. In the revised manuscript, Figure 3 has been redrawn to enhance readability. Besides, we also uploaded separate high-definition figures in the system. We hope that the corrections will meet with approval.

Comments:

  1. in table 1: what does VIP mean? Also, please give in full the word "exptl",

Responses:

   Thanks for the reviewer’s kind remind. “VIP” represents “the variable importance of projection of related metabolites in the OPLS-DA model” and “exptl” means “experimental m/z". We have added the full name of abbreviations in the table legend.

Comments:

  1. figure 4: fonts are again very small,

Responses:

    Thanks for the reviewer’s kind remind. We have enlarged the font in the Figure 4, please see in the revised manuscript.

Comments:

  1. figure 5: please give in the legend the physical meaning of red and blue colours,

Responses:

    Thanks for the reviewer’s kind remind. We have added the descriptions about colors in the revised manuscript.

    Specifically, as follows: [ corr = correlation. Red represents positive correlation and blue represents negative correlation. The deeper the color, the greater the correlation.]

Comments:

  1. in discussion, after line 583, the authors are invited to include one more paragraph highlighting the main outcomes of their work in relation to composition changes of gut microflora and relate these changes to possible anti-inflammatory effects.

Responses:

    Thanks for the reviewer’s kind remind. The comment is very helpful to improve the quality of discussion part. We have added discussions highlighting the main outcomes of our work in relation to composition changes of gut microflora and relate these changes to possible anti-inflammatory effects in the revised manuscript.

  Specifically, as follows:【There are hundreds of millions of bacterial communities in the intestines of mammals, which are interdependent of their hosts and participate in various physiological activities: metabolism, immunity, and regulation of endocrine and nervous system functions1. Emerging evidence suggests that low-grade inflammation is a marker of metabolic disorders such as obesity, type 2 diabetes, and non-alcoholic fatty liver disease 2 3. These diseases are characterized by changes in the intestinal microbiota and its metabolites, which migrate from the intestinal tract through the damaged intestinal barrier, affecting metabolic organs such as the liver and adipose tissue 4 5. Other studies found that obesity, the propensity to gain weight, dyslipidemia, insulin resistance and low-grade inflammation were more prevalent in subjects exhibiting low gut bacterial richness 6 7. Additionally, it was suggested that certain “proinflammatory” bacterial strains such as Ruminococcus gnavus or Bacteroides species, might dominate, while “anti-inflammatory”’ strains, such as Faecalibacterium prausnitzii, are less prevalent 8. Lactic acid bacteria (LAB) are best known for imparting beneficial health effects. Numerous studies showed that lactic acid bacteria resist oxidation, regulate immune function, reduce cholesterol, promote digestion and prevent cancer. so the inclusion in the diet could be positive 9. By definition, LAB produce large amounts of lactic acid using carbohydrates. Lactococcus, Enterococcus, Pediococcus, Leuconostoc, Lactobacillus, Bifidobacterium, Streptococcus, Vagococcus, Tetragenococcus and Weissella are some lactic acid bacterial genera that have been well studied for their health benefits 10. As a probiotic, Lactobacillus johnsonii plays an important role in maintaining the balance of intestinal flora and regulating the immune system. Studies have found that it promotes the growth and development of animals, reduces inflammatory reactions, prevents diarrhea, increases the number of beneficial bacteria in the intestine, and regulates the balance of intestinal microflora 11. After colonization by Lactobacillus yoelii, the number of CD4+ and CD8+ cells in the small intestine and spleen were the highest. Although less effective than fecal bacterial transplantation, colonization with Escherichia coli and Lactobacillus johnsonii both increased the proportion of regulatory T cells (Tregs) and activated DC and completely restored the intestinal memory/effector T cell population on Day 28. Interestingly, only L. johnsonii recolonization maintained colonic IL-10 production 12. Other studies demonstrated that L. johnsonii increases the level of reduced glutathione in the blood, thus improving the mitochondrial morphology and function of the liver by reducing liver lipids and improving systemic glucose metabolism 13. Weissella cibaria is the dominant heterotypic lactic acid fermentation bacterium found in the early stage of the natural fermentation of pickled cabbage. Due to the complexity of fermentation and the diversity of products, it can produce more flavor substances and give pickled cabbage a better flavor 14. Shashank et al. isolated four new Weiss strains from fermented dosa batter and human infant fecal sample and found that all four tolerated gastric fluid (pH = 3.0) and bile salt, moderate cell surface hydrophobicity, reduced cholesterol, and adhered to CaCo2 cells and gastric mucin. The cell experiment proved that all four strains inhibited production of nitric oxide and IL-6 by a mouse macrophage induced by a lipopolysaccharide and inhibit the production of IL-8 in human epithelial cells 15. In this study, it was found that dietary AAP supplementation can significantly up-regulate the abundance of L. johnsonii and Weissella cibaria, but its health promotion effect based on the regulation of intestinal flora needs to be further explored. 】

Comments:

  1. The role of inflammation is not mentioned at all in the MS. I would invite the authors to include some discussion on inflammation and chronic diseases. Useful references that the authors may wish to consider:  Nutrients | Free Full-Text | Cholesterol versus Inflammation as Cause of Chronic Diseases (mdpi.com) and Nutrients | Free Full-Text | Inflammation, not Cholesterol, Is a Cause of Chronic Disease (mdpi.com)

Responses:

    Thanks for the reviewer’s kind remind and useful references. The comment is very helpful to improve the quality of discussion part. We have added discussions about gut microflora, inflammation and chronic diseases in the revised manuscript.

  Specifically, as follows:【There are hundreds of millions of bacterial communities in the intestines of mammals, which are interdependent of their hosts and participate in various physiological activities: metabolism, immunity, and regulation of endocrine and nervous system functions1. Emerging evidence suggests that low-grade inflammation is a marker of metabolic disorders such as obesity, type 2 diabetes, and non-alcoholic fatty liver disease 2 3. These diseases are characterized by changes in the intestinal microbiota and its metabolites, which migrate from the intestinal tract through the damaged intestinal barrier, affecting metabolic organs such as the liver and adipose tissue 4 5. Other studies found that obesity, the propensity to gain weight, dyslipidemia, insulin resistance and low-grade inflammation were more prevalent in subjects exhibiting low gut bacterial richness 6 7. Additionally, it was suggested that certain “proinflammatory” bacterial strains such as Ruminococcus gnavus or Bacteroides species, might dominate, while “anti-inflammatory”’ strains, such as Faecalibacterium prausnitzii, are less prevalent 8. Lactic acid bacteria (LAB) are best known for imparting beneficial health effects. Numerous studies showed that lactic acid bacteria resist oxidation, regulate immune function, reduce cholesterol, promote digestion and prevent cancer. so the inclusion in the diet could be positive 9. By definition, LAB produce large amounts of lactic acid using carbohydrates. Lactococcus, Enterococcus, Pediococcus, Leuconostoc, Lactobacillus, Bifidobacterium, Streptococcus, Vagococcus, Tetragenococcus and Weissella are some lactic acid bacterial genera that have been well studied for their health benefits 10. As a probiotic, Lactobacillus johnsonii plays an important role in maintaining the balance of intestinal flora and regulating the immune system. Studies have found that it promotes the growth and development of animals, reduces inflammatory reactions, prevents diarrhea, increases the number of beneficial bacteria in the intestine, and regulates the balance of intestinal microflora 11. After colonization by Lactobacillus yoelii, the number of CD4+ and CD8+ cells in the small intestine and spleen were the highest. Although less effective than fecal bacterial transplantation, colonization with Escherichia coli and Lactobacillus johnsonii both increased the proportion of regulatory T cells (Tregs) and activated DC and completely restored the intestinal memory/effector T cell population on Day 28. Interestingly, only L. johnsonii recolonization maintained colonic IL-10 production 12. Other studies demonstrated that L. johnsonii increases the level of reduced glutathione in the blood, thus improving the mitochondrial morphology and function of the liver by reducing liver lipids and improving systemic glucose metabolism 13. Weissella cibaria is the dominant heterotypic lactic acid fermentation bacterium found in the early stage of the natural fermentation of pickled cabbage. Due to the complexity of fermentation and the diversity of products, it can produce more flavor substances and give pickled cabbage a better flavor 14. Shashank et al. isolated four new Weiss strains from fermented dosa batter and human infant fecal sample and found that all four tolerated gastric fluid (pH = 3.0) and bile salt, moderate cell surface hydrophobicity, reduced cholesterol, and adhered to CaCo2 cells and gastric mucin. The cell experiment proved that all four strains inhibited production of nitric oxide and IL-6 by a mouse macrophage induced by a lipopolysaccharide and inhibit the production of IL-8 in human epithelial cells 15. In this study, it was found that dietary AAP supplementation can significantly up-regulate the abundance of L. johnsonii and Weissella cibaria, but its health promotion effect based on the regulation of intestinal flora needs to be further explored. 】

References:

  1. Costa, M. C.; Weese, J. S., Understanding the Intestinal Microbiome in Health and Disease. The Veterinary clinics of North America. Equine practice 2018, 34 (1), 1-12.
  2. Panchal, S. K.; Brown, L., Cholesterol versus Inflammation as Cause of Chronic Diseases. Nutrients 2019, 11 (10).
  3. Tsoupras, A.; Lordan, R.; Zabetakis, I., Inflammation, not Cholesterol, Is a Cause of Chronic Disease. Nutrients 2018, 10 (5).
  4. Al Bander, Z.; Nitert, M. D.; Mousa, A.; Naderpoor, N., The Gut Microbiota and Inflammation: An Overview. International journal of environmental research and public health 2020, 17 (20).
  5. Herbert, T.; Niv, Z.; E, A. T.; Eran, E., The intestinal microbiota fuelling metabolic inflammation. Nature reviews. Immunology 2020, 20 (1).
  6. Le Chatelier, E.; Nielsen, T.; Qin, J.; Prifti, E.; Hildebrand, F.; Falony, G.; Almeida, M.; Arumugam, M.; Batto, J. M.; Kennedy, S.; Leonard, P.; Li, J.; Burgdorf, K.; Grarup, N.; Jorgensen, T.; Brandslund, I.; Nielsen, H. B.; Juncker, A. S.; Bertalan, M.; Levenez, F.; Pons, N.; Rasmussen, S.; Sunagawa, S.; Tap, J.; Tims, S.; Zoetendal, E. G.; Brunak, S.; Clement, K.; Dore, J.; Kleerebezem, M.; Kristiansen, K.; Renault, P.; Sicheritz-Ponten, T.; de Vos, W. M.; Zucker, J. D.; Raes, J.; Hansen, T.; Meta, H. I. T. c.; Bork, P.; Wang, J.; Ehrlich, S. D.; Pedersen, O., Richness of human gut microbiome correlates with metabolic markers. Nature 2013, 500 (7464), 541-6.
  7. Aron-Wisnewsky, J.; Prifti, E.; Belda, E.; Ichou, F.; Kayser, B. D.; Dao, M. C.; Verger, E. O.; Hedjazi, L.; Bouillot, J. L.; Chevallier, J. M.; Pons, N.; Le Chatelier, E.; Levenez, F.; Ehrlich, S. D.; Dore, J.; Zucker, J. D.; Clement, K., Major microbiota dysbiosis in severe obesity: fate after bariatric surgery. Gut 2019, 68 (1), 70-82.
  8. Cotillard, A.; Kennedy, S. P.; Kong, L. C.; Prifti, E.; Pons, N.; Le Chatelier, E.; Almeida, M.; Quinquis, B.; Levenez, F.; Galleron, N.; Gougis, S.; Rizkalla, S.; Batto, J. M.; Renault, P.; consortium, A. N. R. M.; Dore, J.; Zucker, J. D.; Clement, K.; Ehrlich, S. D., Dietary intervention impact on gut microbial gene richness. Nature 2013, 500 (7464), 585-8.
  9. Francesca, D. F.; Edoardo, P.; Danilo, E., The food-gut axis: lactic acid bacteria and their link to food, the gut microbiome and human health. FEMS microbiology reviews 2020, 44 (4).
  10. Masood, M. I.; Qadir, M. I.; Shirazi, J. H.; Khan, I. U., Beneficial effects of lactic acid bacteria on human beings. Critical reviews in microbiology 2011, 37 (1), 91-8.
  11. Xin, J.; Zeng, D.; Wang, H.; Sun, N.; Zhao, Y.; Dan, Y.; Pan, K.; Jing, B.; Ni, X., Probiotic Lactobacillus johnsonii BS15 Promotes Growth Performance, Intestinal Immunity, and Gut Microbiota in Piglets. Probiotics and antimicrobial proteins 2020, 12 (1), 184-193.
  12. Ekmekciu, I.; von Klitzing, E.; Neumann, C.; Bacher, P.; Scheffold, A.; Bereswill, S.; Heimesaat, M. M., Fecal Microbiota Transplantation, Commensal Escherichia coli and Lactobacillus johnsonii Strains Differentially Restore Intestinal and Systemic Adaptive Immune Cell Populations Following Broad-spectrum Antibiotic Treatment. Frontiers in microbiology 2017, 8, 2430.
  13. R., R. R.; Manoj, G.; Zhipeng, L.; Manuel, G. J.; Renee, G.; Christopher, G.; Franziska, B.; Hyekyoung, Y.; W., P. J.; Stephany, V. P.; D., W. K.; Briana, F.; Benjamin, P.; B., J. D.; Giorgio, T.; David, B.; J., S. T.; Amiran, D.; Andrey, M.; Natalia, S., Transkingdom interactions between Lactobacilli and hepatic mitochondria attenuate western diet-induced diabetes. Nat Commun 2021, 12 (1).
  14. Yu, H.-S.; Jang, H. J.; Lee, N.-K.; Paik, H.-D., Evaluation of the probiotic characteristics and prophylactic potential of Weissella cibaria strains isolated from kimchi. LWT 2019, 112.
  15. Shashank, S.; Ruchika, B.; Ankit, S.; Paramdeep, S.; Ramandeep, K.; Pragyanshu, K.; K, P. R.; K, B. R.; Praveen, R.; Padma, A.; Kumar, B. S.; Mahendra, B.; Jaspreet, K.; Kiran, K. K., Probiotic attributes and prevention of LPS-induced pro-inflammatory stress in RAW264.7 macrophages and human intestinal epithelial cell line (Caco-2) by newly isolated Weissella cibaria strains. Food Funct 2018, 9 (2).

    We have tried our best to revise and improve the manuscript and made great changes in the manuscript according to the Reviewers’ good comments. We appreciate for Editor/Reviewers’ warm work earnestly. Thank you very much for your help and the manuscript has been resubmitted to your journal. We look forward to your positive response and thank you for your good comments.

Yours sincerely,

Qian Liu

College of Food Science and Technology, Northwest University, Xi’an 710069, China;

Tel: +86 29 88305208;            Fax: +86 29 88305208;

E-mail: liuqian2017@nwu.edu.cn

Reviewer 2 Report

Title: Effects of Auricularia auricula polysaccharides on gut microbiota and metabolic phenotype in mice

Personally, I find the text interesting. Furthermore, this study presents the basis for promising future studies. However, the manuscript presents a series of drawbacks that need to be corrected. In the following lines, I will explain the main mistakes found.

Line 21-22. For humans or also mice?

Abstract. Revise the use of capital letters.

Line 42. “people's table”. Around the world? It is necessary to be more precise and rigorous with the information that is written. It is a recurring error throughout the manuscript.

The level of English should be improved. In some parts of the text, the writing is so colloquial. E.g., as early as… (line 43).

References are missing throughout the entire manuscript. In the introduction, there are many lines in a row without a single reference. This mistake must be corrected.

The introduction is disorganized. It must be rewritten. In addition, the text does not have a common thread that lets us know as we read what the study is going to deal with. This is essential in any article.

Line 134-138. Rewrite.

Line 161-162. Rewrite. This must be carried out under the ethical criteria of animal experimentation. Weren't these protocols followed?

Line 173. “-80 ºC” instead of “-80 oC”.

Where were all the reagents purchased?

In the material and methods section, many parts of the text are written as a laboratory script, not as an explanation in an article. It is written in the form of orders. This should not be so, correct it (e.g., line 200).

Line 232. There is a lot of blank space before the figure. Space is being wasted. It is necessary to move the figure after the text so that this space is not left blank. also, it is impossible to read the legend of figures a and b. The font size is too small. The same mistake is found in line 249. There is almost a whole blank page! This is not permissible. Correct it.

The size of figure 2 is too small. It is impossible to read most of the images that appear in said figure. Same mistake in figure 3. Please, be more careful with the work. It is a pity that for small details the quality of work lowers so much.

The discussion section is quite extensive, so it would be interesting to add a brief section of conclusions to summarize the progress of the study.

FINAL REMARKS

In my opinion, the authors have carried out a really interesting study, with promising expectations for future research. However, there are some issues that should be improved. The level of English and writing should be substantially improved. Therefore, I am suggesting MAJOR REVISIONS. The study should be improved.

Author Response

Dear Editor and Reviewers,

    Thank you for your helpful comments on our manuscript entitled “Effects of Auricularia auricula polysaccharides on gut microbiota and metabolic phenotype in mice” (foods- 1854627) that helps improve the quality of the study. The manuscript has been polished by MDPI language editing services (https://www.mdpi.com/authors/english), and the revised portions are highlighted in blue in the revised manuscript. The main corrections in the manuscript and the responses to the reviewers’ comments are listed below point by point.

Reviewer 2:

    Personally, I find the text interesting. Furthermore, this study presents the basis for promising future studies. However, the manuscript presents a series of drawbacks that need to be corrected. In the following lines, I will explain the main mistakes found.

Responses:

    Thank you for your encouraging comments. We have studied comments carefully and revised our manuscript seriously. The manuscript has been polished by MDPI language editing services (https://www.mdpi.com/authors/english), and the revised portions are highlighted in blue in the revised manuscript. We hope that the corrections will meet with approval.

Comments:

  1. Line 21-22. For humans or also mice?

Responses:

   Thanks for the reviewer’s kind remind. In this study, metabolomic and intestinal microbiome methods were both used to study the effects of Qinba mountain AAP on the serum metabolites and intestinal flora of normal male mice.

Comments:

  1. Abstract. Revise the use of capital letters.

Responses:

    Thanks for the reviewer’s kind remind. We have corrected the inappropriate capitalization in the abstract. The manuscript has been polished by MDPI language editing services, please see in the revised manuscript.

Comments:

  1. Line 42. “people's table”. Around the world? It is necessary to be more precise and rigorous with the information that is written. It is a recurring error throughout the manuscript.

Responses:

     Thanks for the reviewer’s kind remind. The manuscript has been polished by MDPI language editing services, and we have revised the inappropriate descriptions in the manuscript. The revised portions are highlighted in blue in the revised manuscript. We hope that the corrections will meet with approval.

Comments:

  1. The level of English should be improved. In some parts of the text, the writing is so colloquial. E.g., as early as… (line 43).

Responses:

    Thanks for the reviewer’s kind remind. The manuscript has been polished by MDPI language editing services, and we have revised the inappropriate descriptions in the manuscript. The revised portions are highlighted in blue in the revised manuscript. We hope that the corrections will meet with approval.

Comments:

  1. References are missing throughout the entire manuscript. In the introduction, there are many lines in a row without a single reference. This mistake must be corrected.

Responses:

      Thanks for the reviewer’s kind remind. This article is a research article, which has 43 references and 5 figures in the submitted manuscript. In the revised introduction, we have refined the descriptions, checked and inserted references almost every sentence. We hope that the corrections will meet with approval.

Comments:

  1. The introduction is disorganized. It must be rewritten. In addition, the text does not have a common thread that lets us know as we read what the study is going to deal with. This is essential in any article.

Responses:

    Thanks for the reviewer’s kind remind. We have refined and rewritten the introduction in the revised manuscript.  Besides, we make explanations as follows: Food affects the environment of intestinal microbes, which, in turn, affect the absorption and metabolism of food. As a cooking ingredient, the effect and potential mechanism of Auricularia auricula on a healthy body remain unclear. The purpose of this study is to reveal the regulatory effects of AAP on the overall metabolism and gut microbiota homeostasis of mammals. The introduction has been written according to the clue of "Auricularia auricula-overall metabolism-intestinal flora homeostasis-contents of this study".

Comments:

  1. Line 134-138. Rewrite.

Responses:

    Thanks for the reviewer’s kind remind. We have re-written the sentence in passive voice in the revised manuscript.

    Specifically, as follows: [Thirty grams of Auricularia auricula-judae powder, purchased from Shaanxi Tianmei Green Industry Co., Ltd., and grown in the Qinba Mountains of northwest China, were weighed, and extracted using 100 °C water (solid–liquid ratio 1:50) for 2 h.]

Comments:

  1. Line 161-162. Rewrite. This must be carried out under the ethical criteria of animal experimentation. Weren't these protocols followed?

Responses:

    Thanks for the reviewer’s kind remind. We have revised the inappropriate descriptions in the manuscript. The animal experiment protocol was approved by the Animal Ethics Committee of the Laboratory Animal Center of Northwest University (Approval Code: NWU-AWC-20200401M; Approval Date: 2020. 04. 06), and was carried out in accordance with the "Administrative Regulations on Laboratory Animals" of the National Science and Technology Commission of the People's Republic of China. We have stated these in the manuscript, and we ensure that our research complies with the commonly-accepted '3Rs'.

Comments:

  1. Line 173. “-80 ºC” instead of “-80 oC”.

Responses:

    Thanks for the reviewer’s kind remind. We have changed “-80 oC” to““-80 ºC”, please see in the revised manuscript.

Comments:

  1. Where were all the reagents purchased?

Responses:

    Thanks for the reviewer’s kind remind. We have added the descriptions of the source of the reagent. Specifically, as follows: [Auricularia auricula-judae, purchased from Shaanxi Tianmei Green Industry Co., Ltd. Animal feed, purchased from TROPHIC Animal Feed High-Tech Co., Ltd., Nantong, China. Phusion ® High-Fidelity PCR Master Mix with GC Buffer, purchased from New England Biolabs. Methanol, LC-MS grade, purchased from CNW Technologies. Acetonitrile, LC-MS grade, purchased from CNW Technologies. Ammonium acetate, LC-MS grade, purchased from Sigma-Aldrich. Ammonium hydroxide, LC-MS grade, purchased from Fisher Chemical. And other reagents are domestic analytical pure.]

Comments:

  1. In the material and methods section, many parts of the text are written as a laboratory script, not as an explanation in an article. It is written in the form of orders. This should not be so, correct it (e.g., line 200).

Responses:

    Thanks for the reviewer’s kind remind. The manuscript has been polished by MDPI language editing services, and we have revised the inappropriate descriptions in the material and methods section. The revised portions are highlighted in blue in the revised manuscript. We hope that the corrections will meet with approval.

Comments:

  1. Line 232. There is a lot of blank space before the figure. Space is being wasted. It is necessary to move the figure after the text so that this space is not left blank. also, it is impossible to read the legend of figures a and b. The font size is too small. The same mistake is found in line 249. There is almost a whole blank page! This is not permissible. Correct it.

Responses:

    Thanks for the reviewer’s kind remind. We have re-arranged the Figures in the manuscript to avoid a lot of blanks. We have enlarged the font in the all the Figures to enhance readability. We hope that the corrections will meet with approval.

Comments:

  1. The size of figure 2 is too small. It is impossible to read most of the images that appear in said figure. Same mistake in figure 3. Please, be more careful with the work. It is a pity that for small details the quality of work lowers so much.

Responses:

    Thanks for the reviewer’s kind remind. In the revised manuscript, all the figures have been redrawn, especially, the font were enlarged to enhance readability. Besides, we also uploaded separate high-definition figures in the system. We hope that the corrections will meet with approval.

Comments:

  1. The discussion section is quite extensive, so it would be interesting to add a brief section of conclusions to summarize the progress of the study.

Responses:

    Thanks for the reviewer’s kind remind. The comment is very helpful to improve the quality of our study. We have added a brief section of conclusions in the revised manuscript.

    Specifically, as follows: 【This study showed that non-digestible AAP promoted the growth of Lactobacillus johnsonii, Weissella cibaria, Kosakonia cowanii, Enterococcus faecalis, Bifidobacterium animalis and Bacteroides uniformis in normal mice intestines. After AAP dietary supplementation, 51 differential metabolites were found, which were mainly enriched in arginine biosynthesis pathways, followed by arginine and proline metabolism; glycine, serine and threonine metabolism; glycerophospholipid metabolism; and the sphingolipid metabolism pathway. Furthermore, it was found that there is a close correlation between differential bacteria and differential metabolites, which may contribute to the bioactivity of AAP. These results laid the theoretical basis for the further development and use of Auricularia auricula resources from the Qinba Mountains and nutritional foods with AAP as the main functional component.】

      We have tried our best to revise and improve the manuscript and made great changes in the manuscript according to the Reviewers’ good comments. We appreciate for Editor/Reviewers’ warm work earnestly. Thank you very much for your help and the manuscript has been resubmitted to your journal. We look forward to your positive response and thank you for your good comments.

Yours sincerely,

Qian Liu

College of Food Science and Technology, Northwest University, Xi’an 710069, China;

Tel: +86 29 88305208;            Fax: +86 29 88305208;

E-mail: liuqian2017@nwu.edu.cn

Round 2

Reviewer 1 Report

Please, make sure all references mentioned in your reply to the reviewer are included in the MS.

Reviewer 2 Report

Authors have done all what it was requested.

It can be published.